# Stimulating the hippocampal posterior-medial network enhances task-dependent connectivity and memory

Kristen N Warren*, Molly S Hermiller, Aneesha S Nilakantan, Joel L Voss

Interdepartmental Neuroscience Program, Department of Medical Social Sciences, Ken and Ruth Davee Department of Neurology, and Department of Psychiatry and Behavioral Sciences, Feinberg School of Medicine, Northwestern University, Chicago, United States

**Abstract** Successful episodic memory involves dynamic increases in activity across distributed hippocampal networks, including the posterior-medial (PMN) and the anterior-temporal (ATN) networks. We tested whether this up-regulation of functional connectivity during memory processing can be enhanced within hippocampal networks by noninvasive stimulation, and whether such task-dependent connectivity enhancement predicts memory improvement. Participants received stimulation targeting the PMN or an out-of-network control location. We compared the effects of stimulation on fMRI connectivity during an autobiographical retrieval task versus during rest within the PMN and the ATN. PMN-targeted stimulation significantly increased connectivity during autobiographical retrieval versus rest within the PMN. This effect was not observed in the ATN, or in either network following control stimulation. Task-dependent increases in connectivity within the medial temporal lobe predicted improved performance of a separate episodic memory test. It is therefore possible to enhance the task-dependent regulation of hippocampal network connectivity that supports memory processing using noninvasive stimulation.
DOI: https://doi.org/10.7554/eLife.49458.001

*For correspondence:
kristenwarren2015@u.
northwestern.edu

Competing interests: The authors declare that no competing interests exist.

## Introduction

There is substantial interest in treating memory disorders via brain stimulation (*Suthana and Fried, 2014*; *Kim et al., 2016*; *Sreekumar et al., 2017*). Episodic memory depends on the hippocampus (*Scoville and Milner, 1957*; *Squire, 1992*) as well as on the distributed set of regions that form a hippocampal-cortical network (*Buckner et al., 2008*; *Rugg and Vilberg, 2013*; *Eichenbaum, 2000*; *Ranganath et al., 2005*; *Ranganath and Ritchey, 2012*) comprising distinct anterior-temporal and posterior-medial components (*Ranganath and Ritchey, 2012*; *Libby et al., 2012*). The goal of this study was to determine whether noninvasive stimulation targeting the hippocampal-cortical network can enhance network connectivity measured during memory processing, and whether such enhancement is related to episodic memory improvement. We targeted specific portions of the posterior-medial network (PMN) and therefore further hypothesized that stimulation would disproportionately impact the PMN rather than the anterior-temporal network (ATN). This question is of substantial mechanistic and practical significance, given that increased connectivity due to stimulation should manifest primarily during memory processing that depends on the network, and such task-dependent modulation would be essential for any effective intervention to improve memory ability.

Invasive stimulation of the hippocampus or its direct mesial-temporal afferents has primarily been associated with memory disruption (*Jacobs et al., 2016*; *Goyal et al., 2018*; *Kucewicz et al., 2018a*; *Mohan, 2019*). However, hippocampal network regions can serve as a target for memory improvement (*Mohan, 2019*; *Ezzyat et al., 2018*; *Kucewicz et al., 2018b*; *Miller et al., 2015*;

*Shirvalkar et al., 2010*). Noninvasive transcranial electromagnetic stimulation (TMS) targeting cortical PMN locations has been shown to improve memory (*Tambini et al., 2018*; *Hermiller et al., 2019*; *Kim et al., 2018*; *Wang et al., 2014*; *Nilakantan et al., 2019*; *Nilakantan et al., 2017*) and alter hippocampal-cortical network fMRI activity (*Tambini et al., 2018*; *Hermiller et al., 2019*; *Kim et al., 2018*; *Wang et al., 2014*; *Nilakantan et al., 2019*), especially within the PMN (*Kim et al., 2018*; *Wang et al., 2014*; *Nilakantan et al., 2019*), for durations that substantially outlast the stimulation period. Because PMN-targeted stimulation increases connectivity among PMN regions and relatively low hippocampal-cortical network connectivity is associated with poor episodic memory (*Beason-Held, 2011*; *Grady et al., 2006*; *Lustig et al., 2003*; *Damoiseaux et al., 2008*; *Froudist-Walsh et al., 2018*; *Henson et al., 2016*), it is tempting to hypothesize that memory improvements due to stimulation occur via overall increased connectivity of hippocampal-cortical networks. However, successful memory typically relies on dynamic reconfiguration of fMRI connectivity within the PMN and ATN in response to memory demands (*Warren et al., 2018*; *Gratton et al., 2016*; *Kim and Voss, 2019*; *Westphal et al., 2017*; *King et al., 2015*; *Geib et al., 2017*). Thus, stimulation might need to produce task-dependent and location-specific, rather than nonspecific, increases in hippocampal-cortical network connectivity in order to benefit memory.

Several lines of evidence suggest that memory enhancement should require that hippocampal-cortical network fMRI connectivity increases occur in response to memory processing demands. For instance, effective memory encoding and retrieval is associated with location-specific and task-dependent increases of stimulus-evoked fMRI activity within the network (*Paller and Wagner, 2002*; *Paller, 1997*; *Wagner et al., 1998*; *Buckner et al., 2001*; *Kirchhoff et al., 2000*). Correspondingly, better memory is predicted by fMRI connectivity related to specific memory processes, such as recollection (*Kim and Voss, 2019*; *King et al., 2015*). In rodents, theta-gamma synchrony in the hippocampus is observed during memory processing, indicating connectivity changes modulated by memory demands (*Shirvalkar et al., 2010*). Pharmacological disruption of hippocampal theta-gamma synchrony and memory can be rescued by theta-burst stimulation of the fornix, indicating that this marker of connectivity is critical for memory (*Shirvalkar et al., 2010*). Furthermore, amnestic states, such as those caused by hippocampal lesions, are associated not only with reduced hippocampal-cortical connectivity, but also with increased connectivity between the hippocampal network and other networks (*Froudist-Walsh et al., 2018*; *Henson et al., 2016*), suggesting that nonspecific increases in connectivity can be problematic. Demonstration that noninvasive stimulation can affect memory task-dependent changes in connectivity within specific targeted portions of the hippocampal-cortical network therefore is crucial for its evaluation as an effective memory intervention. This question has not been addressed, as the current standard for assessing the effects of brain stimulation on networks is via fMRI connectivity measured during the resting state (*Wang et al., 2014*; *Canals et al., 2009*; *Eldaief et al., 2011*; *Fox et al., 2012*), which by definition is insensitive to task-dependent connectivity.

We evaluated whether stimulation can alter task-dependent fMRI connectivity within the PMN and ATN. One group of participants (n = 16) received multi-session, high-frequency TMS targeting the PMN via parietal cortex, which is a robust component of the PMN (*Ranganath and Ritchey, 2012*; *Libby et al., 2012*). A separate control group (n = 16) received the same TMS regimen targeting a control prefrontal cortex (PFC) location that is not robustly part of the PMN. Each group also received site-specific sham-control stimulation, delivered at a subthreshold intensity which should not influence brain function, administered in counterbalanced order with real stimulation (*Figure 1*). Because both real and sham stimulation were delivered in different groups (targeting the PMN versus the control PFC location), this design guarded against influences from any nonspecific or subjective differences between real and sham stimulation, which would be similar for both stimulation locations. We compared the effects of stimulation on fMRI connectivity within the PMN, ATN, and whole-brain (via an exploratory analysis) measured 24 hr later during a memory retrieval task versus during the resting state. The memory retrieval task involved extended periods of autobiographical memory retrieval, which has been shown to cause robust hippocampal-cortical network connectivity changes primarily in PMN regions compared to the resting state (*Warren et al., 2018*). Thus, we were able to test for task-dependent (autobiographical retrieval versus rest) and network-specific (PMN-targeted versus PFC control) effects of stimulation on fMRI connectivity.

We hypothesized that PMN-targeted stimulation would increase fMRI connectivity during memory retrieval as compared to the resting state relative to sham-control stimulation, whereas out-of-

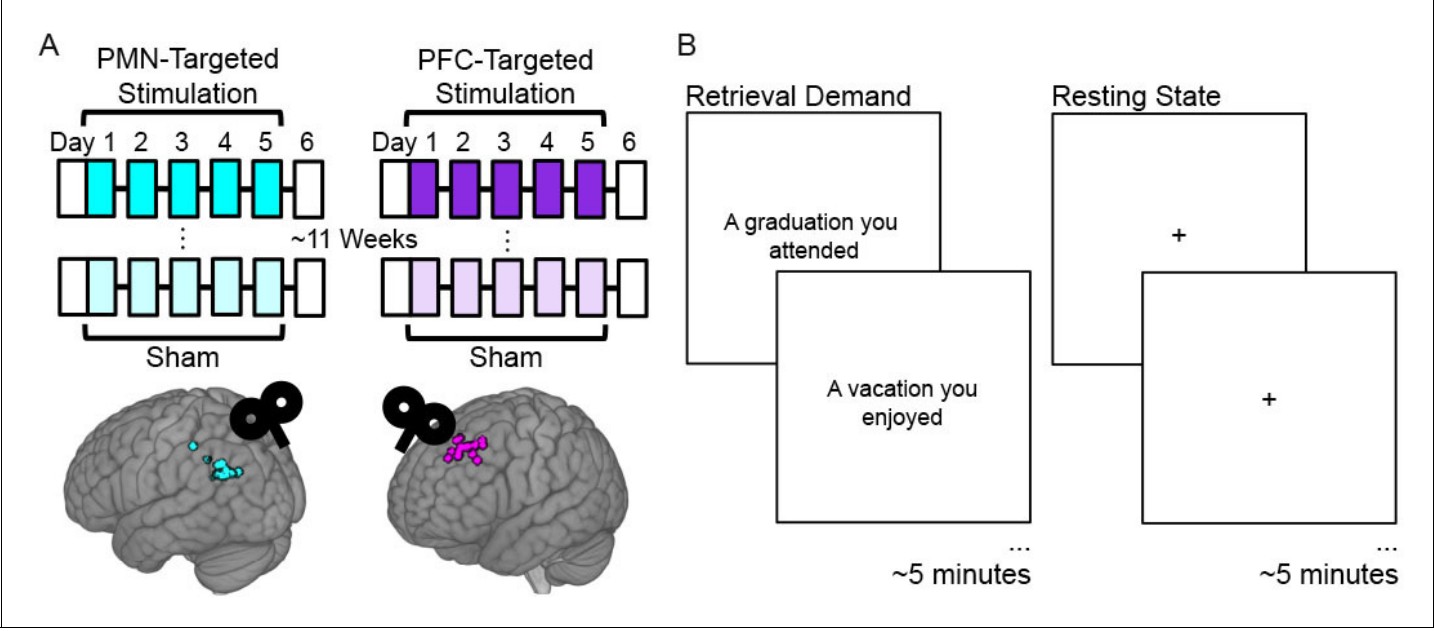

**Figure 1.** Experiment Design. (**A**) Subjects received five consecutive daily sessions of high-frequency (20 Hz) repetitive TMS delivered to a subject-specific parietal cortex location of the PMN selected based on high resting-state fMRI connectivity with the hippocampus (PMN-targeted Stimulation). Subjects received real stimulation and sham stimulation during different weeks, in counterbalanced order with an average 11.5 week washout period between these conditions. Before and ~ 24 hr after stimulation, subjects completed fMRI and memory assessments (white boxes). The same procedures were performed for a distinct control group of subjects, but with stimulation delivered to subject-specific locations of out-of-network prefrontal cortex (PFC-Targeted Stimulation). Circles indicate stimulation locations for each participant. (**B**) fMRI connectivity was measured during the resting state and during an autobiographical memory retrieval task, for which subjects were shown prompts describing common life events and asked to vividly recall personal events matching the prompts.

DOI: https://doi.org/10.7554/eLife.49458.002

network PFC-targeted stimulation would not. We additionally hypothesized that these connectivity changes would be specific to the PMN, as it (i) was targeted, (ii) contributes to autobiographical memory retrieval (*Ranganath and Ritchey, 2012*), and (iii) responds to the stimulation regimen that was used (*Kim et al., 2018*; *Wang et al., 2014*; *Nilakantan et al., 2019*). We therefore assessed whether the task-dependent modulation of connectivity due to PMN-targeted stimulation occurred specifically for the PMN versus the ATN, using a priori defined regions of interest for each network (*Libby et al., 2012*). Furthermore, we tested whether the positive influence of PMN-targeted stimulation on the up-regulation of fMRI connectivity during memory retrieval would predict episodic memory improvement measured in an independent task. This allowed us to address the hypothesis that selective increases in fMRI connectivity during memory retrieval serves as an indicator of effective hippocampal network function that can be modulated by PMN-targeted TMS.

## Results

### Stimulation effects on task-dependent fMRI connectivity within hippocampal networks

We examined the task-dependent effects for PMN-targeted versus PFC-targeted control stimulation on network-wide interconnectivity (i.e., mean connectivity of each region to all other regions) in the PMN and the ATN using a linear mixed effects model with factors stimulation condition (stimulation versus sham), task (retrieval versus rest), and stimulation location (PMN-targeted versus PFC-targeted), controlling for scan data quality (tSNR) (see Materials and methods: Data Analysis). The primary hypothesis was that there would be three-way interaction within the PMN among stimulation condition, task, and stimulation target, reflecting increased fMRI interconnectivity due to stimulation for autobiographical retrieval versus rest, and selectively for PMN-targeted stimulation versus out-of-

network PFC-targeted control stimulation. Indeed, stimulation effects on PMN interconnectivity were greater during retrieval versus rest following PMN-targeted stimulation relative to PFC-targeted stimulation (3-way interaction: T(89)=2.80, p=0.006) (*Figure 2A*). This task-sensitive relative interconnectivity increase for PMN-targeted versus PFC-targeted stimulation was not found for the ATN (3-way interaction: T(89)=1.48, p=0.14) (*Figure 2A*). Other main effects and interactions were nonsignificant in both networks (T < 2, *Supplementary file 1*).

As shown in *Figure 2A*, PMN-targeted stimulation, relative to sham, significantly increased task-dependent connectivity during retrieval versus rest (interaction of stimulation condition and task) in the PMN (T(45)=2.02, p=0.049) but led to a numeric but non-significant increase in the ATN (T(45) =0.60, p=0.55). In contrast, task-dependent connectivity decreased non-significantly in both networks for the group of subjects receiving PFC-targeted stimulation (PMN: T(45)=-1.60, p=0.12; ATN: T(45)=-1.06, p=0.29). Thus, although the primary statistical comparisons were performed using the logic of the sham condition (within-subjects) and control stimulation location (between groups) experimental design, the same pattern of results was evident when examining each stimulation group independently.

We next identified regions that were driving these stimulation effects on network-wide task-dependent interconnectivity. Correlation matrices for the PMN and ATN were constructed, with the same linear mixed effects model used for each pair of regions as in the whole-network analysis to test the task-dependent effects on their connectivity following PMN-targeted versus PFC-targeted control stimulation. The pairwise connections showing significantly greater task-dependent stimulation effects during retrieval following PMN-targeted stimulation following correction for multiple comparisons (*Figure 2B*) included higher-order visual regions of the PMN, especially precuneus and lingual gyrus, with the hippocampus. Notably, hippocampal connectivity with these regions has been particularly implicated in episodic recollection (*Rugg and Vilberg, 2013*; *King et al., 2015*). We also computed average connectivity of each region to all other within-network regions to identify those regions that increased in task-dependent connectivity with the entire rest of the network. Within the PMN, the same regions showing pairwise task-dependent connectivity increases (hippocampus, precuneus, and lingual gyrus) also exhibited significant increases in task-dependent connectivity with the rest of the network, after correcting for multiple comparisons (*Figure 2C*). No ATN regions showed these effects.

## Increased whole-brain fMRI connectivity during memory retrieval

Task-dependent effects of stimulation on fMRI connectivity were measured via exploratory whole-brain analyses (*Warren et al., 2018*; *Gotts et al., 2012*; *Cole et al., 2010*) of data from resting-state and retrieval task scans to allow for identification of regions unbiased by a priori designations of regions of interest. To validate the retrieval scan as an assay for memory-related connectivity and to show that our method for whole-brain fMRI connectivity measurement is sensitive to changes caused by memory retrieval, we first assessed the main effect of task (retrieval versus rest) independent from the factors stimulation type (stimulation versus sham) and stimulation location (PMN-targeted versus PFC-targeted) via linear mixed effects modeling (See Materials and methods: Data Analysis). There was a main effect of task in the PMN and the broader hippocampal-cortical network as well as regions typically associated with autobiographical memory, with greater connectivity during the retrieval task than during rest (*Figure 3—figure supplement 2*). This replicates our previous findings using this analysis method and retrieval task, and is consistent with many findings of fMRI connectivity increases due to similar memory retrieval tasks (*Svoboda et al., 2006*; *Bellana et al., 2017*; *McCormick et al., 2015*; *Rabin et al., 2010*). Therefore, our methods detect valid indicators of task-dependent changes in functional connectivity associated with memory.

## Exploratory whole-brain analysis of task-dependent effects of PMN-targeted stimulation

In order to evaluate the selectivity of effects of stimulation on the PMN, we next examined the task-dependent effects of stimulation using an exploratory, voxel-wise, whole-brain connectivity analysis approach (*Warren et al., 2018*; *Gotts et al., 2012*; *Cole et al., 2010*). We used the same linear mixed effects model as in the network analysis for the three-way interaction of condition, task, and stimulation location. Consistent with the targeted network analysis, there were many regions that

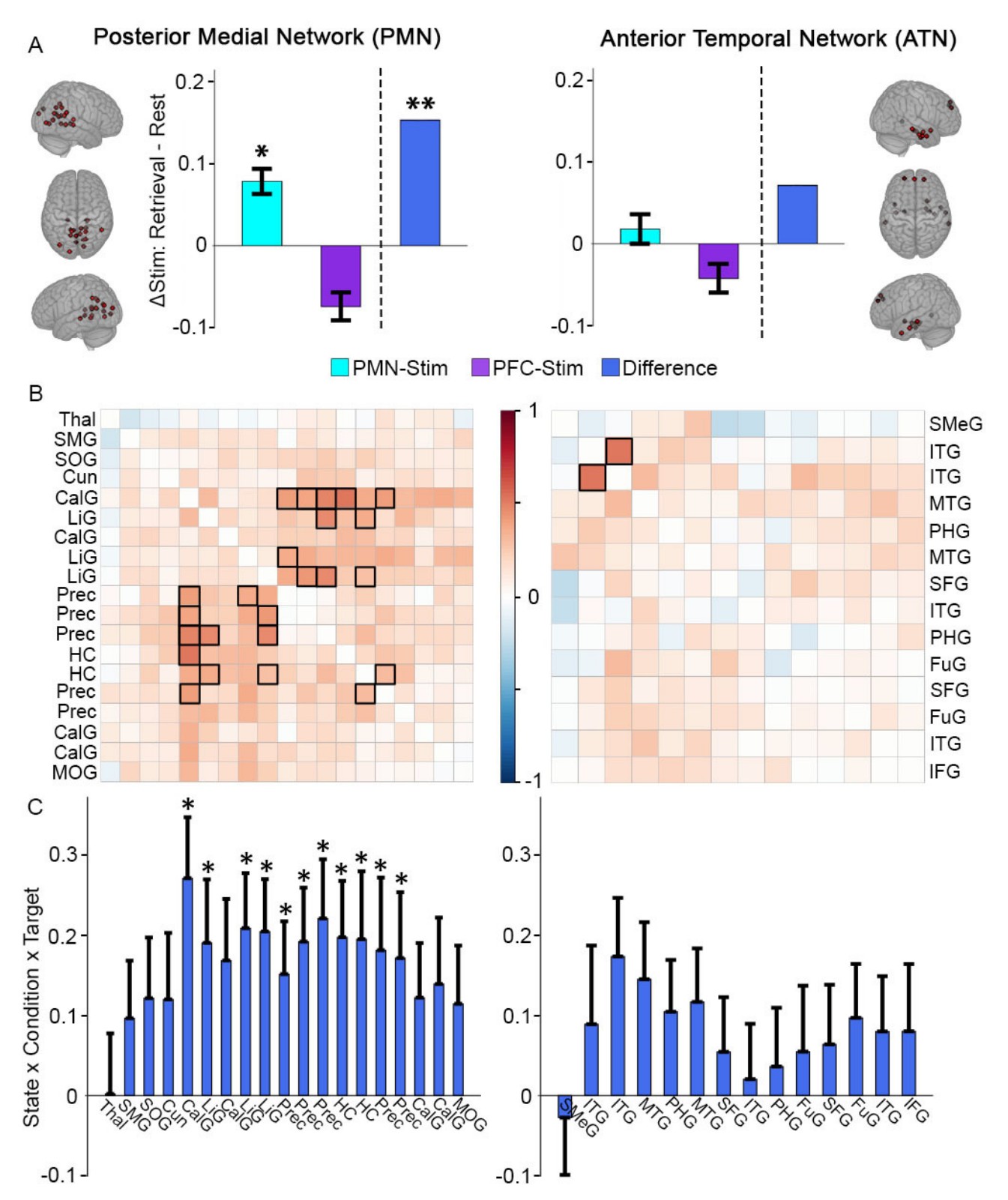

**Figure 2.** Greater memory task-dependent connectivity increases due to PMN-targeted versus PFC-targeted control stimulation for the PMN. (A) Stimulation effect on mean connectivity (all regions to all other regions) during memory retrieval relative to rest for the PMN (left) and the ATN (right). Error bars indicate subject-level standard error of the mean. *=p < 0.05, **=p < 0.01 for the interaction between stimulation condition and state in each group, or the three-way interaction between condition, state, and stimulation location. (B) The same effect of stimulation on connectivity between each

*Figure 2 continued on next page*

*Figure 2 continued*

pair of regions in the PMN (left) or ATN (right). Coloration represents the beta-weight of the three-way interaction effect, with significant cells shown in bold (FDR corrected p<0.05). (**C**) The same effect of stimulation on connectivity for each region in the PMN to all other PMN regions (left) and each region in the ATN to all other ATN regions (right). Error bars indicate standard error of the mean of the three-way interaction. *=FDR corrected p<0.05. Regions are shown in *Figure 2—figure supplement 1* and region abbreviations are expanded in *Supplementary file 2*.

DOI: https://doi.org/10.7554/eLife.49458.003

The following source data and figure supplement are available for figure 2:

**Source data 1.** PMN/ATN network-based analysis.

DOI: https://doi.org/10.7554/eLife.49458.005

**Figure supplement 1.** Network regions of interest.

DOI: https://doi.org/10.7554/eLife.49458.004

exhibited significantly greater stimulation effects on whole-brain connectivity during retrieval versus rest following PMN-targeted stimulation relative to PFC-targeted control stimulation (*Figure 3*).

Because of the difference identified across stimulation groups, we next tested the interaction of stimulation condition (stimulation versus sham) and task (retrieval versus rest) using linear mixed effects models performed separately for each stimulation group (PMN-targeted or PFC-targeted; See Materials and methods: Data Analysis). For the PMN-targeted stimulation group, there was significant interaction of stimulation by task in many regions particularly within the PMN (*Figure 4A*). All but one area showed relatively greater connectivity for stimulation relative to sham during retrieval compared to rest, thereby demonstrating the hypothesized task-dependent increase in fMRI connectivity due to stimulation independently in the group of subjects receiving PMN-targeted stimulation (*Figure 4C*).

The majority of regions (93.5%) demonstrating this interaction effect overlapped with the network that was sensitive to autobiographical retrieval (*Figure 3—figure supplement 2*), indicating that task-dependent connectivity changes due to stimulation occurred for regions that contribute to memory retrieval. Furthermore, 61% of regions demonstrating the interaction effect were within a priori defined hippocampal-cortical network locations (8 PMN, 2 ATN, and 1 DMN, see Materials and methods), especially in the PMN, with the remaining falling among anterior salience, sensorimotor, precuneus, and auditory networks that have been associated with autobiographical retrieval (*Svoboda et al., 2006*; *Bellana et al., 2017*; *McCormick et al., 2015*; *Rabin et al., 2010*). Thus, PMN-targeted stimulation had memory task-dependent effects primarily within hippocampal-cortical regions of a priori interest, with other areas showing this effect being involved generally in memory retrieval.

PFC-targeted control stimulation caused a nearly opposite pattern of fMRI connectivity change relative to PMN-targeted stimulation. As was the case for PMN-targeted stimulation, there was significant interaction of stimulation by task on connectivity (*Figure 4B*). However, all regions showed the opposite direction of connectivity changes relative to PMN-targeted stimulation, with greater connectivity during rest than during retrieval following PFC-targeted stimulation relative to sham (*Figure 4C*). Notably, the area of prefrontal cortex stimulated in the control condition does not overlap with either the PMN or the ATN and participates in a variety of non-memory cognitive operations such as attention and maintenance of external awareness (*Van Calster et al., 2017*; *Chou et al., 2017*). Stimulation of the control location may thus have caused a relative increase in these operations during rest and/or a disruption of memory-related processing during the retrieval task.

The locations of this interaction effect were consistent with results obtained from the three-way interaction model (*Figure 3*), which overlapped with 94.4% of the areas obtained via the PMN-targeted stimulation results and 86.7% of the PFC-targeted stimulation results. We then used a whole-brain analysis approach to thoroughly compare spatial distributions for all regions that contributed to connectivity effects in the group-level analyses (See Materials and methods). Regions driving the task-dependent connectivity increases due to stimulation were categorized as belonging to PMN versus ATN (*Ranganath et al., 2005*; *Ranganath and Ritchey, 2012*). As in the targeted analysis of effects on PMN and ATN (*Figure 2*), PMN-targeted stimulation produced memory task-dependent increases in connectivity in 31 PMN regions and only 5 ATN regions. In contrast, PFC-targeted stimulation produced memory task-dependent decreases in an evenly distributed set of PMN and ATN regions (12 PMN regions, 13 ATN regions). There was a significant difference in the relative

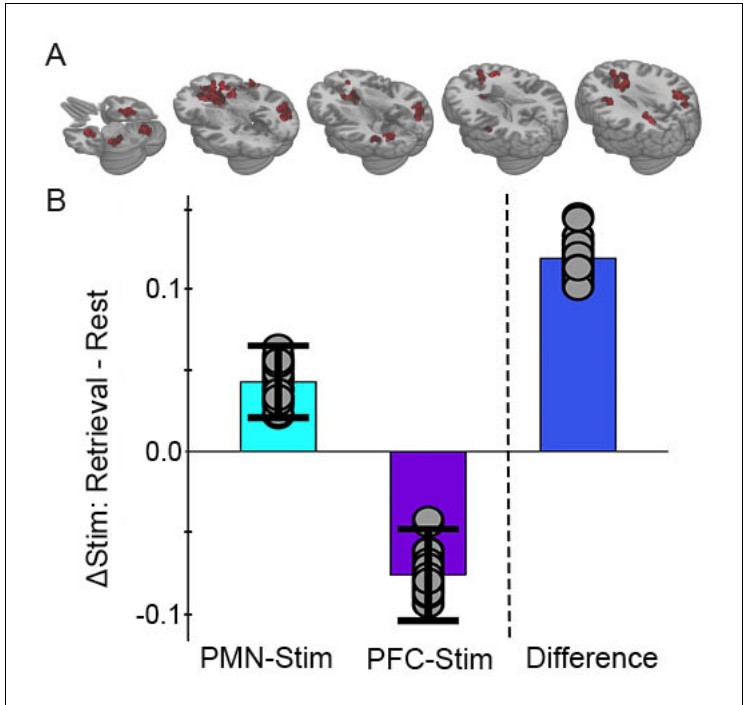

**Figure 3.** Greater memory task-dependent connectivity increases due to PMN-targeted versus prefrontal-control stimulation. **(A)** Regions showing a significant interaction between condition, task, and group, with red coloration indicating stimulation increased connectivity more during retrieval than during rest in the PMN-targeted group relative to the prefrontal-control group. All regions showed greater memory task-related connectivity change following PMN-targeted stimulation. Comprehensive view shown in Figure Supplement. **(B)** Mean stimulation effect on connectivity during memory retrieval relative to rest for all supra-threshold regions. Error bars are provided for illustrative purposes and indicate subject-level standard error of the mean for supra-threshold regions. Points indicate the mean effect for each supra-threshold region. Note: Statistical values are not indicated, as this would be redundant with the statistical definition of these supra-threshold regions.

DOI: https://doi.org/10.7554/eLife.49458.006

The following source data and figure supplements are available for figure 3:

**Source data 1.** Whole-brain analysis, full model.
DOI: https://doi.org/10.7554/eLife.49458.010
**Figure supplement 1.** Selective effects of stimulation on memory-specific connectivity.
DOI: https://doi.org/10.7554/eLife.49458.007
**Figure supplement 2.** Increased fMRI connectivity during memory retrieval.
DOI: https://doi.org/10.7554/eLife.49458.008
**Figure supplement 3.** Common effects of stimulation across sites.
DOI: https://doi.org/10.7554/eLife.49458.009

distribution of task-dependent effects (irrespective of directionality) on PMN versus ATN regions in the PMN-targeted relative to PFC-targeted stimulation conditions (Yates-corrected $X^2(1)=8.56$, p=0.003). Thus, the relatively selective effects of stimulation on PMN were consistent across the targeted analysis of PMN and ATN regions of interest and the whole-brain analyses.

## Increased memory task-dependent connectivity predicts episodic memory improvement

Episodic memory improvement was measured using an independent task (i.e., separate from the autobiographical retrieval and resting-state tasks used to assess task-dependent stimulation effects on fMRI connectedness). This task involved item recognition and context recollection (see Materials and methods: Memory Task) (*Figure 5A*). Because context recollection is more heavily dependent on the PMN than item recognition (*Ranganath and Ritchey, 2012*; *Libby et al., 2012*)

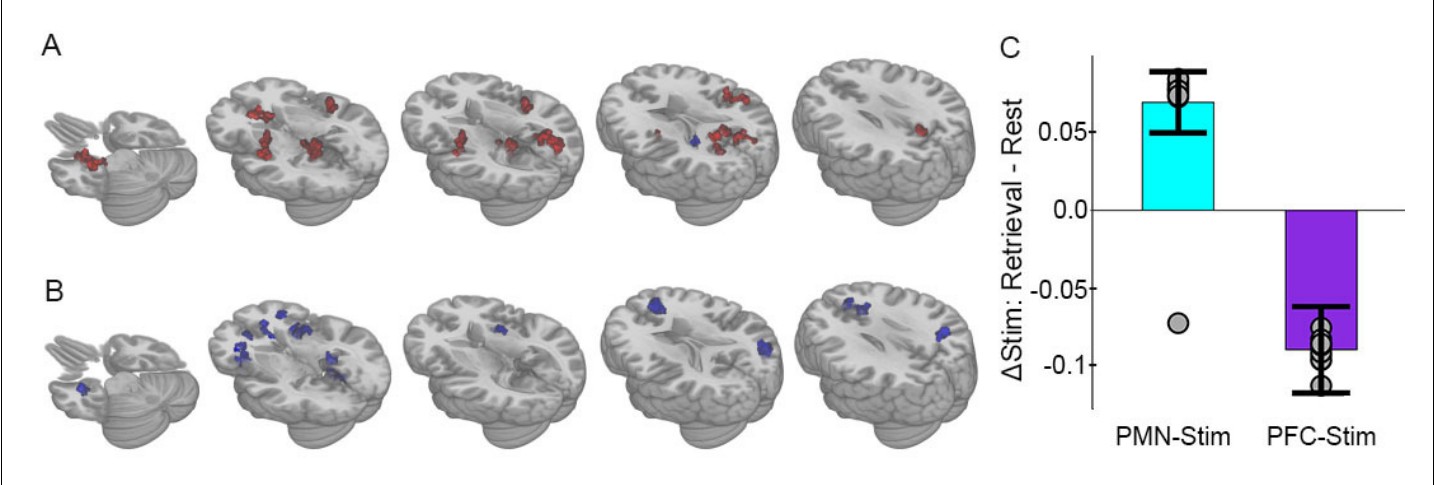

**Figure 4.** Selective effects of PMN-targeted stimulation on memory-related connectivity. (**A**) Regions showing significant interaction between stimulation condition and cognitive task following PMN-targeted stimulation, with red coloration indicating stimulation increased connectivity more during retrieval than during rest and blue coloration indicating the opposite effect. Comprehensive view shown in Figure Supplements. (**B**) Regions showing significant interaction between stimulation condition and cognitive task following PFC-targeted control stimulation. (**C**) Mean stimulation effect on connectivity during memory retrieval relative to rest for all supra-threshold regions. Error bars are provided for illustrative purposes and indicate subject-level standard error of the mean for all supra-threshold regions. Points indicate the mean effect for each supra-threshold region. Note: Statistical values are not indicated, as this would be redundant with the statistical definition of these supra-threshold regions.

DOI: https://doi.org/10.7554/eLife.49458.011

The following source data and figure supplements are available for figure 4:

**Source data 1.** Whole-brain analysis, PMN-targeted group.
DOI: https://doi.org/10.7554/eLife.49458.014
**Source data 2.** Whole-brain analysis, PFC-targeted group.
DOI: https://doi.org/10.7554/eLife.49458.015
**Figure supplement 1.** Selective effects of stimulation on memory-specific connectivity.
DOI: https://doi.org/10.7554/eLife.49458.012
**Figure supplement 2.** Selective effects of stimulation on memory-specific connectivity.
DOI: https://doi.org/10.7554/eLife.49458.013

and based on our previous findings (*Kim et al., 2018*; *Wang et al., 2014*; *Nilakantan et al., 2019*; *Nilakantan et al., 2017*), we predicted that PMN-targeted stimulation would improve context recollection selectively (*Ranganath and Ritchey, 2012*). Considering both stimulation groups, there were no baseline differences in memory performance before stimulation was delivered (pre-stimulation vs pre-sham: T(31)=-1.23, p=0.23). Consistent with the previously reported results in 30 of the 32 subjects analyzed here (*Kim et al., 2018*), PMN-targeted stimulation improved context recollection (post-stimulation vs post-sham: T(15)=1.93, p=0.036) whereas PFC-targeted control stimulation did not (post-stimulation vs post-sham: T(15)=-0.09, p=0.46), with significantly greater improvement for PMN-targeted stimulation relative to PFC-targeted stimulation (T(15)=1.88, p=0.04). Stimulation had no effect on item recognition for either stimulation location (PMN-targeted: T(15)=0.94, p=0.36; PFC-targeted: T(15)=-1.34, p=0.20).

We next tested whether task-dependent increases in fMRI connectivity predicted memory improvement. Because of the critical role of the medial temporal lobe in memory (*Tambini et al., 2010*; *Kim et al., 2018*) and on our previous findings showing left-lateralized effects of the same stimulation parameters on hippocampal fMRI activity (*Wang et al., 2014*; *Nilakantan et al., 2019*), we focused our analysis on a location in the left medial temporal lobe which showed task-dependent stimulation effects in each group (*Figure 5B*). The amount that PMN-targeted stimulation caused task-dependent increases in connectivity for retrieval versus rest was associated with greater improvement in context recollection memory (robust regression F(15) = 6.64, $r^2$ = 0.12, p=0.022). Although PFC-targeted stimulation was associated with net opposite task-dependent effects as PMN-targeted stimulation (i.e., greater connectivity increases for rest compared to retrieval), the

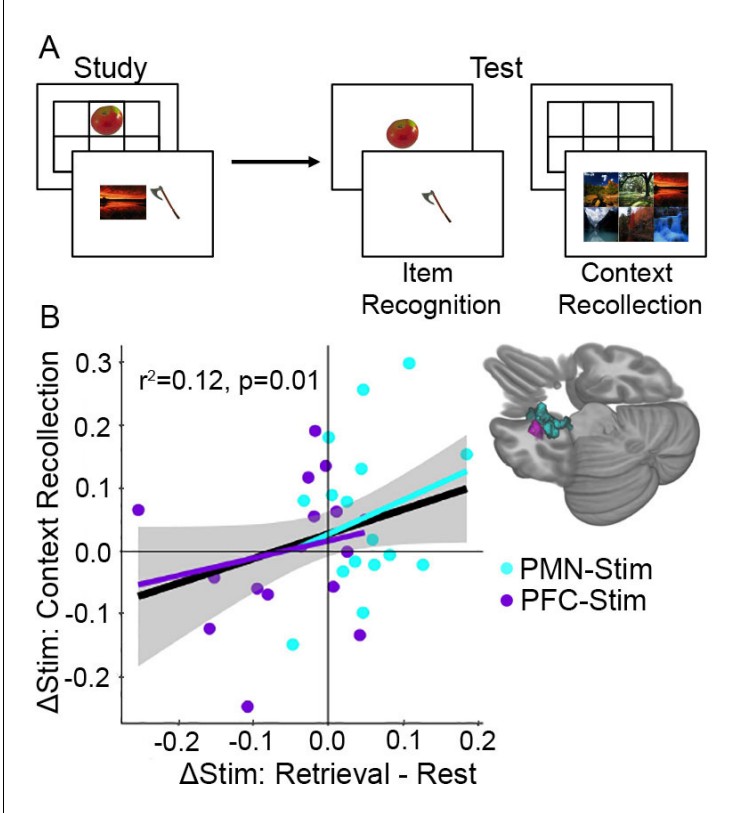

**Figure 5.** Increased memory-dependent connectivity predicts episodic memory improvement. (**A**) Episodic memory task design. Participants studied trial-unique objects paired with either scene or location contexts. After a delay, we assessed object recognition memory and contextual recollection memory. (**B**) Connectivity values were pooled across the MTL regions identified in each group showing an interaction between stimulation condition and cognitive task. Scatterplot shows the relationship between each subject's change in retrieval task connectivity relative to rest and their change in context recollection performance following stimulation. Greater specificity of connectivity change to the memory task in the left MTL was associated with improvement in context recollection in each group independently (cyan and purple regression lines) as well as collectively across all 32 subjects (black regression line).

DOI: https://doi.org/10.7554/eLife.49458.016

same positive relationship was identified between memory task-dependent connectivity and context recollection as for PMN-targeted stimulation. That is, relative increases in connectivity during retrieval than during rest predicted relative improvement in source recollection (robust regression F(15)=11.14, $r^2$ = 0.16, p=0.005). Based on the similarity of these effects, we pooled connectivity values across the two regions of interest and found that, irrespective of stimulation condition, the degree to which stimulation increased task-dependent connectivity during retrieval compared to rest predicted the amount of context recollection improvement (F(31) = 6.71, $r^2$ = 0.12, p=0.01) (*Figure 5*). Thus, memory task-dependent connectivity increases for the left medial temporal lobe is a robust indicator of episodic memory improvement due to stimulation.

## Discussion

These findings demonstrate memory task-dependent increases in the expression of fMRI connectivity changes caused by PMN-targeted stimulation. These effects were selective to a priori defined regions of the PMN relative to the ATN, which was confirmed via exploratory whole-brain analysis. Task-dependent connectivity increases in these regions were also specific to the PMN-targeted stimulation condition, relative to PFC-targeted control stimulation. Furthermore, retrieval-related connectivity increases in left medial temporal cortex and hippocampus due to stimulation predicted

context recollection improvement measured in a separate task. Thus, enhancing hippocampal connectivity during memory processing is functionally critical and is achievable via noninvasive stimulation. Furthermore, these task-dependent effects were measured ~ 24 hr after stimulation, indicating that stimulation led to upregulation of connectivity during a subsequent memory task administered long after, and without any specific relationship to, stimulation delivery.

The PMN and ATN are functionally distinct components of the larger hippocampal-cortical network and are thought to differentially support memory processing related to context recollection versus item recognition, respectively (*Ranganath and Ritchey, 2012*; *Libby et al., 2012*). Here, we report that increased connectivity due to stimulation was selective for PMN regions and for the autobiographical memory retrieval task. This selectivity is consistent with the role of the PMN in autobiographical memory retrieval (*Ranganath and Ritchey, 2012*) as well as with the stimulation location, which was within the PMN (*Libby et al., 2012*). These findings are consonant with our previous report that the same stimulation protocol increased stimulus-evoked activity during the encoding of item-context pairings selectively within the PMN (*Kim et al., 2018*; *Nilakantan et al., 2019*), and further supports hypothesized functional distinctions of the PMN and ATN. Future studies could seek stronger evidence for functional distinction by attempting to target the ATN with the goal of identifying crossover interaction of the effects of ATN-targeted versus PMN-targeted stimulation.

Connectivity changes due to stimulation were measured during a memory retrieval task (autobiographical retrieval task) versus rest, yet they predicted the effects of stimulation on memory ability measured using an independent item/source episodic memory test. Although autobiographical retrieval and episodic memory are superficially distinct, there is accumulating evidence that they are supported by similar cognitive operations and brain regions (*McCormick et al., 2015*; *Andrews-Hanna et al., 2014*; *Burianova et al., 2010*; *Daselaar et al., 2008*). Our findings of a relationship of stimulation effects on connectivity during autobiographical retrieval with episodic memory performance underscores this similarity. Indeed, we have previously found that the PMN-targeted stimulation parameters used here improve a variety of episodic memory tasks of different formats, including face-word paired associates (*Wang et al., 2014*), item-scene paired associates (*Kim et al., 2018*; *Nilakantan et al., 2019*), item-location associations (*Kim et al., 2018*; *Nilakantan et al., 2019*), and precise spatial recall (*Nilakantan et al., 2017*). Although we did not assess stimulation effects on the success of autobiographical memory retrieval, the current findings emphasize that stimulation targeting the PMN improves a variety of memory measures. Though these memory tasks are typically considered distinct (paired associate versus spatial recall, episodic versus autobiographical), they all have been associated with the PMN (*Buckner et al., 2008*; *Svoboda et al., 2006*; *McCormick et al., 2015*; *Andrews-Hanna et al., 2014*; *Burianova et al., 2010*; *Daselaar et al., 2008*) and therefore all likely respond to PMN-targeted stimulation.

Although resting-state fMRI connectivity has proven useful in characterizing the effects of stimulation on brain network function, including to understand memory improvements due to stimulation (*Tambini et al., 2018*; *Hermiller et al., 2019*; *Wang et al., 2014*; *Kim and Voss, 2019*), the current findings highlight one of the many limitations of resting-state fMRI for this purpose. That is, relationships between resting-state connectivity and network function are not well specified. Here, there was strong modulation of the effects of stimulation on connectivity based on whether it was measured during a memory retrieval task versus during rest, indicating that resting-state fMRI alone was an incomplete assay for stimulation effects. Furthermore, the functional relevance of stimulation for memory performance was related to its effect on the task-dependent change in connectivity, indicating that resting-state fMRI alone would not have identified functionally relevant effects of stimulation on connectivity.

It is likely that individuals engage in a variety of uncontrolled and typically unmeasured cognitive operations during resting-state fMRI and that these at least partially drive resting-state fMRI connectivity outcomes (*Van Calster et al., 2017*; *Chou et al., 2017*; *Kucyi et al., 2018*; *Hurlburt et al., 2015*). Our finding that stimulation effects on connectivity are particularly strong during a memory retrieval task is especially problematic for resting-state fMRI as an outcome for memory interventions given that subjects frequently and variably retrieve memories during resting-state fMRI (*Van Calster et al., 2017*; *Chou et al., 2017*; *Kucyi et al., 2018*; *Hurlburt et al., 2015*). Variable effects of stimulation on fMRI connectivity at rest would therefore be expected based on the content and quality of memory retrieval subjects experience during scanning. The current experiment accounted for these challenges by giving an explicit retrieval task during connectivity measurement, which permitted

differentiation of the PMN-targeted versus PFC-targeted stimulation conditions and identified functionally relevant effects of stimulation on task-dependent connectivity related to memory performance.

As our main comparison was between autobiographical memory retrieval and rest, our interpretation is limited without a different cognitive task control. It is therefore possible that similar effects of stimulation on task-dependent connectivity could result using tasks other than those that involve memory. However, different cognitive demands have been strongly associated with connectivity of distinct functional networks (*Cole et al., 2014*; *Rissman et al., 2004*; *Mennes et al., 2013*), with PMN connectivity primarily associated with episodic and autobiographical memory (*Buckner et al., 2008*; *Svoboda et al., 2006*; *McCormick et al., 2015*; *Andrews-Hanna et al., 2014*; *Burianova et al., 2010*; *Daselaar et al., 2008*). It is therefore reasonable to propose that task-dependent effects of stimulation on PMN connectivity would likely be specific for those that involve memory demands, including but not limited to autobiographical retrieval. Notably, whereas PMN-targeted stimulation increased PMN connectivity during retrieval more so than during rest, PFC-targeted stimulation demonstrated the opposite pattern (rest > retrieval). Because the PMN but not the dorsolateral prefrontal cortex has generally been associated with vivid autobiographical memory retrieval (*Svoboda et al., 2006*; *St Jacques et al., 2011*), this finding suggests that task-dependent connectivity increases occurred for the task that matched the function of the stimulated network. Future studies should directly investigate task-specificity using multiple task conditions, with an emphasis on determining whether memory demands are required to generate results such as those reported here.

We used an area of dorsolateral prefrontal cortex as a control stimulation location as it is distinct from the PMN. Indeed, counter to the effects of PMN-targeted stimulation, PFC-targeted stimulation caused nonsignificant reductions of PMN task-dependent connectivity (*Figure 2A*) and significant reductions of task-dependent connectivity in several distributed regions independent from the PMN (*Figure 4B*). These findings are consistent with our previous report showing reductions in task-evoked activity during memory encoding in prefrontal cortex due to the PFC-targeted stimulation protocol used here (*Kim et al., 2018*). Although resting-state connectivity of dorsolateral prefrontal cortex with the hippocampus is not robust (*Kahn et al., 2008*; *Yeo et al., 2011*), some analyses incorporating task-based connectivity have identified areas of dorsolateral prefrontal cortex that align with the ATN (*Libby et al., 2012*). However, these areas are distinct from the dorsolateral prefrontal location that we stimulated and, indeed, PFC-targeted stimulation caused no significant change in ATN task-dependent connectivity. Thus, PFC-targeted stimulation was a suitable control for PMN-targeted stimulation, based on a priori considerations and on the distinct pattern of findings that was identified for PFC-targeted stimulation versus PMN-targeted stimulation. An important direction for future research is to determine whether the multi-day stimulation regimen used in the current experiment applied to dorsolateral prefrontal cortex can improve cognitive functions thought to depend on this area.

Although stimulation that influenced task-dependent PMN connectivity (PMN-targeted stimulation) was delivered via lateral parietal cortex, the most robust effects on the PMN occurred in medial occipital-parietal and hippocampal portions of the PMN. This is consistent with our previous findings obtained with the current multi-day stimulation protocol, whereby the most robust effects on connectivity and memory-related activity due to lateral parietal stimulation occurred for the hippocampus and medial occipital-parietal regions rather than the stimulation site (*Kim et al., 2018*; *Wang et al., 2014*; *Nilakantan et al., 2019*). One possibility is that although lateral parietal cortex provides anatomical projections to the mesial temporal lobe (*Mesulam et al., 1977*; *Pandya and Seltzer, 1982*) by which its stimulation can influence hippocampal activity, it has relatively weak functional connectivity relative to medial occipital-parietal portions of the network (*Kahn et al., 2008*; *Yeo et al., 2011*) and therefore is less influenced by the influence of stimulation on hippocampal neuroplasticity. Indeed, we have previously found that the degree to which a region's connectivity increases due to stimulation is predicted by its baseline connectivity with the hippocampus (*Wang et al., 2014*), and this relationship has been replicated in an independent sample (*Freedberg et al., 2019*).

An alternative possibility is that the effects of brain stimulation are locally disruptive at the stimulated location (*Jacobs et al., 2016*; *Goyal et al., 2018*; *Kucewicz et al., 2018a*; *Mohan, 2019*) but can generate facilitation in connected regions (*Mohan, 2019*; *Ezzyat et al., 2018*; *Kucewicz et al.,*

*2018b*; *Miller et al., 2015*; *Shirvalkar et al., 2010*). Indeed, like PMN-targeted stimulation, the effects of PFC-targeted stimulation on task-dependent connectivity also occurred for distributed regions other than the stimulated location (*Figure 4B*), but without involvement of the hippocampus. Future research will be needed to adjudicate among alternative possibilities for how network-targeted brain stimulation influences the hippocampal network as well as other neurocognitive networks.

To summarize, our findings suggest that PMN-targeted brain stimulation increases activity coupling among PMN regions when these regions are engaged by memory processing demands, rather than nonspecifically engaged during rest. Furthermore, task-dependent connectivity increases in the medial temporal lobe predicted improvement in a separate memory task. Thus, memory enhancement by brain stimulation relies on dynamic (i.e., activity-dependent) rather than static changes in connectivity among select portions of the hippocampal-cortical network, thereby reflecting a form of effective rather than functional connectivity (*Friston, 2011*). The relationship between task-dependent connectivity and memory improvement was robust for the left hippocampus and surrounding medial temporal cortex, consistent with its established role in memory encoding and retrieval (*Scoville and Milner, 1957*; *Squire, 1992*; *Ranganath and Ritchey, 2012*). This finding is important given that nonspecific increases in connectivity could be detrimental to memory as well as other cognitive abilities, as this would entail less-selective participation of the hippocampal-cortical network in memory (*Shirvalkar et al., 2010*; *Froudist-Walsh et al., 2018*; *Henson et al., 2016*; *Warren et al., 2018*; *Gratton et al., 2016*; *Kim and Voss, 2019*; *Westphal et al., 2017*; *King et al., 2015*; *Geib et al., 2017*). Stimulation-based interventions for memory disorders should therefore strive to achieve memory task-dependent functional engagement, as identified here using PMN-targeted noninvasive stimulation.

## Materials and methods

### Participants

Thirty-two adults participated in the experiment (22 females, mean age = 25.6 years, range = 18–34). Data from two additional participants were collected but discarded due to excessive motion (see below). All conditions of interest were fully counterbalanced in the final sample contributing data to analyses. All participants had normal or corrected-to-normal vision and did not report a history of neurological or psychiatric disorders or current drug use. Participants were eligible for MRI and TMS procedures according to standard MRI and TMS safety-screening questionnaires. Eligibility contraindications were evaluated by a neurologist (S.V.) Memory performance data from 30/32 subjects contributing to the analysis of the relationship between connectivity and memory (see below) has been previously reported (*Kim et al., 2018*). The Institutional Review Board at Northwestern University provided approval for this study. All participants gave written, informed consent and were paid for their participation.

### Experiment design

Participants completed a 2 week experiment involving one week of full-intensity stimulation and one week of sham stimulation (*Figure 1A*) to either a posterior medial network target in the left lateral parietal lobe (PMN-targeted stimulation, N = 16) or a control site in the left lateral prefrontal cortex (PFC-targeted stimulation, N = 16). The order of these two weeks was counterbalanced, and the first day of each week was separated by a delay of at least 4 weeks (mean interval = 11.52 weeks, range = 4.71–37.14 weeks). About 2 hr before receiving stimulation on the first day of each week and ~ 24 hr after five consecutive daily stimulation sessions (mean delay = 23.3, SD = 2.50 hr from the final stimulation session), participants completed a resting-state scan, an autobiographical memory retrieval task scan, and a task-based fMRI memory paradigm (*Figure 1B*, *Figure 4A*). Task-associated behavioral and fMRI data have been reported elsewhere for 30 of the 32 subjects, with two subjects replaced due to excessive motion during resting-state scans to achieve the full sample reported here (N = 32). The present analyses focus on post-stimulation versus post-sham comparisons, as in several previous experiments (*Kim et al., 2018*; *Nilakantan et al., 2019*; *Nilakantan et al., 2017*).

## Resting-State and autobiographical memory retrieval task fMRI scans

The resting-state and the autobiographical memory retrieval task scans used the same EPI sequence, which each lasted 5.5 min. During the resting-state scan, a fixation cross was presented continuously, and participants were instructed to remain awake with eyes open and fixated on the cross. During the retrieval task scan, participants were shown text prompts for common life events, such as 'A graduation you attended' or 'A vacation you enjoyed,' and were instructed to vividly recall these events (*Figure 1C*) (*Warren et al., 2018*). Each prompt was shown for 20 s with a 10 s inter-prompt interval, and participants were told to imagine the event for the full 30 s period. Ten event prompts were shown consecutively during the scan. There were six sets of prompts each including distinct events. Each subject received a different version at each assessment, with order counterbalanced across subjects receiving full stimulation and sham stimulation during the first week.

## Imaging acquisition and processing

Participants were scanned using a Siemens MAGNETOM Prisma scanner with a 64-channel head/neck coil. The resting-state and retrieval task scans were both acquired using a T2* weighted echo planar imaging (EPI) sequence (TR = 555 ms, TE = 22 ms, multiband factor = 8, isotropic voxel size = 2×2 × 2 mm, FOV = 208×192 mm, flip angle = 47˚, volumes = 550). Structural imaging was acquired using MPRAGE T1-weighted scans (TR = 2170 ms, TE = 1.69 ms, voxel size = 1 mm$^3$, FOV = 25.6 cm, flip angle = 7˚, 176 sagittal slices).

Data were preprocessed using AFNI (Version AFNI_16.1.15), with the same processing steps used for retrieval task and resting-state scans. The first five EPI volumes were removed to avoid intensity normalization artifact. AFNI's *3dDespike* was used to remove large transient volumes. The remaining volumes were slice-time and multiband corrected. Six estimated motion parameters (x, y, z, roll, pitch, and yaw) were estimated using 3dvolreg. Data from two additional participants were excluded for high motion (17% and 13% of TRs censored at FD threshold 0.3 mm;<5% censored for all included subjects). For included participants, pairwise comparisons were made for post-stim, post-sham, retrieval task, and resting-state scan framewise displacement (FD) and temporal signal to noise ratio (tSNR) in the PMN-targeted and PFC-targeted control stimulation groups. All FD comparisons were nonsignificant (p>0.05), however, marginal differences in temporal signal to noise ratio (tSNR) were found between post-stimulation and post-sham retrieval scans in the PMN-targeted stimulation group (T(15)=-2.18, p=0.05), thus all statistical analyses used tSNR as a covariate of no interest. Volumes were co-registered to the anatomical scan and then transformed into standardized Talairach and Tournoux space (TT_N27 atlas). Images were then smoothed with an isotropic 4.0 mm full-width half-maximum Gaussian kernel and masked using AFNI's *3dAutomask*. The six motion parameters estimated earlier were then regressed out of the timeseries using AFNI's *3dDeconvolve*, after which a bandpass filter was applied (0.01–0.1 Hz) using AFNI's *3dTproject*. A group mask was created by merging masks of all subjects using AFNI's *3dmerge* and including only voxels that were present in the final preprocessed datasets in all subjects.

## Identification of stimulation locations

Individualized left lateral parietal (PMN-targeted) or left prefrontal (PFC-targeted control) stimulation locations were determined based on high resting-state fMRI connectivity with a left hippocampal seed using a procedure previously described (*Kim et al., 2018*; *Wang et al., 2014*; *Nilakantan et al., 2017*). Briefly, resting state data from the first visit was used to select a hippocampal volume of interest for each participant by identifying a voxel in the body of the left hippocampus closest to MNI [−29,–25, −13] (mean distance = 2.51 mm, range = 0.00–6.71 mm) for which fMRI connectivity was maximal to contralateral hippocampus. This location was used for seed-based connectivity analysis (AFNI's *InstaCorr*) using a seed radius of 2 mm.

For the PMN-targeted stimulation group, the stimulation location was selected as the peak connectivity cluster within left lateral parietal cortex, within an anatomical mask of angular and supramarginal gyri and inferior parietal lobule close to MNI [−47,–68, 36] (mean distance = 10.2 mm, range = 0.0–35.8 mm from this coordinate; *Figure 1B*). For the PFC-targeted stimulation group, we generated a functional mask of left dorsolateral prefrontal cortex through Neurosynth (*Yarkoni et al., 2011*) (on April 28, 2016) as meta-analytic co-activation with the left hippocampus (MNI [−29,–25, −13]). The logic of this selection is that we intended to stimulate an area of the

dorsolateral prefrontal cortex that participates in memory processing along with the hippocampus but that does not have significant functional connectivity with the hippocampus. Indeed, seed-based functional connectivity of the indicated left hippocampal location in Neurosynth does not identify the left dorsolateral prefrontal cortex, which is consistent with other evidence that this area has relatively weak functional connectivity with the hippocampus (*Kahn et al., 2008*; *Yeo et al., 2011*; *Mesulam et al., 1977*; *Pandya and Seltzer, 1982*). To roughly match the targeting approach used in the PMN-targeted stimulation group, the stimulation location was selected as the peak resting-state functional connectivity cluster within this mask close to MNI [−23, 40, 43] (mean distance = 10.5 mm, range = 0.0–19.9 mm; *Figure 1B*). Thus, functional connectivity was used to select the exact stimulation location for both conditions, but only for PMN-Stim could this site be selected solely based on functional connectivity with the hippocampus, whereas meta-analytic co-activation was needed to provide an approximate area for the PFC-Stim condition. The stimulation target was transformed for each participant to original space for anatomically guided stimulation. The same stimulation location was used for each subject for both stimulation and sham weeks.

## Transcranial Magnetic Stimulation (TMS)

The MagPro X100 system with a MagPro Cool-B65 liquid-cooled butterfly coil was used (MagVenture A/S, Farum, Denmark). A frameless stereotactic system (Localite GmbH, St. Augustin, Germany) used individual MRIs for anatomical targeting of stimulation and for recording coil locations relative to the brain for each TMS pulse. Resting motor threshold (MT) was determined visually based on the minimum stimulator output required to generate a contraction of the *abductor pollicis brevis* for 5 out of 10 consecutive single pulses. Repetitive TMS was planned at 100% MT for each day of the stimulation week and at 10% MT for each day of the sham week, although these values were lowered due to discomfort for five subjects in the PMN-targeted stimulation group (to 95%, 90%, 82%, 80%, and 72% MT) and for three subjects in the PFC-targeted group (to 89%, 83%, and 74% MT). The final mean stimulator output intensity for full stimulation in the PMN-targeted group was 52.5 (SD = 8.5) and 51.2 (SD = 7.7) for the PFC-targeted group (T(31)=0.22, p=0.83). The mean output for sham was 5.42 (SD = 0.90) in the PMN-targeted group and 5.19 (SD = 0.66) for the PFC-targeted group. Although subjects could easily discriminate real stimulation and sham conditions based on intensity, they were unaware of hypotheses regarding the effects of higher-intensity stimulation on memory. Furthermore, subjects in both the PMN-targeted and PFC-targeted stimulation groups would have been equally aware of the difference between stimulation and sham conditions, whereas we hypothesized that effects of stimulation on memory-related connectivity would result for the PMN-targeted stimulation group only. Each daily TMS session consisted of 40 consecutive trains of 20 Hz pulses for 2 s followed by 28 s of no stimulation (1600 pulses per session, 20 min total).

## fMRI data analysis

Two components of the hippocampal-cortical network, the PMN and the ATN, were of particular interest, as the stimulation location was within the PMN and we have previously shown that the effects of this same stimulation protocol on task-based fMRI activity during memory encoding are greater for regions within the PMN network than the ATN (*Kim et al., 2018*). Regions in the PMN and ATN were defined a priori based on previous studies of fMRI connectivity with parahippocampal and perirhinal cortex, respectively (*Libby et al., 2012*; *Ritchey et al., 2014*). Network regions were 6-mm-radius spheres centered on the peak coordinates of each network location (*Figure 2—figure supplement 1*). The spatially averaged time series for each region was extracted from each subject's post-stimulation and post-sham retrieval and resting-state scans and used to construct a correlation matrix for each state and condition (R package *corrplot*, RStudio 1.1.453). Network connectivity was first assessed as the average of all pairwise correlations within a network (*Figure 2A*). This was compared among conditions using a linear mixed effects model (R packages *lme4* and *lmerTest*) with the factors stimulation condition (stimulation/sham), cognitive task (retrieval/rest), stimulation group (PMN-targeted/PFC-targeted), their three-way interaction, and a covariate tSNR term (see Image Acquisition and Processing). Linear mixed effects modelling was used rather than more traditional repeated-measures ANOVA analysis in order to permit inclusion of covariates of no interest (tSNR). To verify that the order of the real stimulation and sham conditions (which was counterbalanced) did not influence task-dependent connectivity, we performed as a control the same analysis but

including condition order as an additional factor. There was no significant main effect of order in either the PMN or the ATN (PMN: T(83)=-1.6, p=0.12; ATN:T(83)=−0.44, p=0.66) and no significant interaction of order with stimulation condition in either network of interest (PMN: T(83)=1.29, p=0.20; ATN:T(83)=−0.30, p=0.77).

To identify which connections contributed to the average network effects, we used the same three-way interaction model to identify effects on connectivity for each pair of regions (*Figure 2B*). Significant links were defined by a two-tailed pair-wise f-value threshold of p<0.05 after FDR correction. Finally, we identified which regions showed the greatest task-dependent stimulation effects by averaging each region's correlation with all other network regions and comparing these values with the three-way interaction model (*Figure 2C*). Significant regions were defined by a two-tailed voxel-wise f-value threshold of p<0.05 after FDR correction.

Next, we expanded our analysis to examine the effects of stimulation on whole-brain fMRI connectivity. Following our previous study examining retrieval task and resting-state differences (*Warren et al., 2018*), we used a whole-brain global connectedness analysis to identify the differential effects of stimulation on resting-state and retrieval fMRI connectivity. Global fMRI connectedness maps were created for each participant's post-stimulation and post-sham rest and retrieval fMRI scans separately (*Warren et al., 2018*; *Gotts et al., 2012*; *Cole et al., 2010*). The correlation of each voxel's timeseries was computed against every other voxel within the brain mask and the mean correlation with all other voxels was stored back into the voxel (AFNI's *3dTcorrMap*), giving a measure of how correlated each voxel is with all other voxels throughout the entire brain. These global fMRI connectedness maps were transformed using Fisher's z to create normally distributed values. This method is data-driven yet conservative, as any significant correlations must survive being washed out by weaker or opposite-direction correlations in other voxels throughout the entire brain. Regions identified in using this data-driven approach were then codified based on membership to well-characterized functional networks to aid interpretation.

Mean connectivity was compared among conditions using linear mixed effects models using AFNI's *3dLME*. We first used the same model as in the PMN and ATN ROI analyses, which included the factors stimulation condition (stimulation/sham), cognitive task (retrieval/rest), stimulation group (PMN-targeted/PFC-targeted), their three-way interaction, and a control tSNR term (see Image Acquisition and Processing) (*Figure 3*). We then created two models which independently examined PMN-targeted and PFC-targeted group results with the factors stimulation condition, cognitive task, their interaction, and tSNR (*Figure 4*). A gray matter mask was then applied to exclude any regions falling in white matter, created by averaging the MPRAGE scans of all 32 subjects, which was then used to create the gray matter mask using AFNI's *3dSeg*. Significant clusters were defined by a two-tailed voxel-wise f-value threshold of p<0.05, which is typical for fMRI connectedness given that experimental effects of subsets of voxels are averaged with null effects for the majority of the brain (*Warren et al., 2018*; *Gotts et al., 2012*; *Cole et al., 2010*) and that two-tailed testing avoids inflation of false positive results present in the majority of neuroimaging experiments using one-tailed testing (*Chen, 2018*; *Cox et al., 2017*; *Eklund et al., 2016*). We controlled false positives by computing a threshold for the minimum number of contiguous supra-threshold voxels using permutation testing. Permutation testing was conducted by running the three-factor model 1000 times with random flipping of factor labels. A probability distribution of cluster sizes was generated across all permutations for each factor using the two-tailed f-value threshold of p<0.05 from our primary analysis. Cluster size cutoffs were then defined as the size with only a 5% probability of finding any cluster that size or larger given random factor label assignment. This identified a threshold of 38 voxels for the stimulation main effect, 36 voxels for the cognitive task main effect, and 36 voxels for the three-way interaction effect. We applied the most stringent of these thresholds, 38 voxels, to all effects to identify significant results. On average, supra-threshold clusters were 2.2 times as large as this threshold (83.3 voxels).

Based on previous work (*Kim et al., 2018*; *Wang et al., 2014*), we had a priori hypothesis that changes in the medial temporal lobe and the episodic memory network (*Ranganath and Ritchey, 2012*) would be particularly related to changes in memory performance following PMN-targeted stimulation. We therefore characterized the network allegiance of the global correlation regions and their 'drivers', the locations which showed changes in connectivity with the global correlation regions and thus were contributing to the effect. To visualize the drivers, regions identified by the interaction effect in the independent group models (PMN-targeted or PFC-targeted) were used as seeds for

voxel-wise whole-brain fMRI correlation analysis. The spatially averaged time series for each region was extracted from each subject's post-stimulation and post-sham retrieval and resting-state scans and correlated with the time series of every other voxel in the brain mask. Seed-based correlation maps across subjects were then compared using the same two-factor linear model as in the global correlation step (AFNI's *3dLME*). Significant clusters were defined by a voxel-wise f-value threshold of p<0.001 two-tailed and a cluster extent threshold of 20 contiguous voxels in the interaction effect. This is a highly conservative threshold for all recent thresholding recommendations (*Chen, 2018*; *Cox et al., 2017*; *Eklund et al., 2016*). We again examined the relationship between these effects and the PMN and ATN ROI effects first identified. Clusters from the whole-brain analysis which overlapped with one of the network regions were considered as clusters within that network. Network allegiance of all remaining clusters was identified using an atlas of 14 resting-state functional networks identified in Shirer, et al., 2012 (*Shirer et al., 2012*).

## Episodic memory assessment and analysis of its relationship to fMRI connectivity

The memory task completed post-stimulation and post-sham involved study-test blocks and used two stimulus formats (*Figure 5A*), as described in *Kim et al. (2018)*. For each block, participants studied 42 trial-unique objects either paired with one of six scenes or displayed in one of six locations on a grid, and then memory was tested after a 2 min delay. During test blocks, half of trials were old (studied) objects and half of trials were new (unstudied) objects. Participants first categorized object as 'old' or 'new' and simultaneously rated confidence as 'certain' or 'uncertain' using four response options, providing a measure of item recognition memory. All studied objects were then tested for contextual recollection memory, whereby participants selected the scene or the location associated with the object during the study phase. Behavioral data have been reported previously (*Kim et al., 2018*) for 30 of the 32 subjects included in the present study (i.e., there are only two new memory-task datasets for the current experiment, for the two subjects that were needed to replace those previously excluded due to poor resting-state fMRI data quality), showing effects of stimulation, relative to sham, specifically on context recollection. Therefore, the effects of stimulation on memory performance reported below are largely redundant with the *Kim et al. (2018)* report and not considered novel evidence for stimulation effects on memory accuracy. Instead, these data are included here for the analysis of the relationship between stimulation effects on fMRI connectivity and context recollection accuracy.

At each assessment, item recognition memory was assessed as the proportion of total trials that were correctly recognized as old (hits) or new (correct rejections). Contextual recollection accuracy was assessed as the proportion of correct responses to the original scene or location context (one of six options) given that the object was correctly recognized with high confidence ('certain' response). Based on previous work with these data (*Kim et al., 2018*) and a priori hypothesis of memory improvement due to parietal stimulation, as already reported in *Kim et al. (2018)*, directional (one-tailed) paired t-tests comparing post-stimulation versus post-sham performance were used to verify that stimulation effects on context recollection remained using the two replacement subjects in the current analysis. To reiterate, the goal of including these behavioral data was not to demonstrate the effects of stimulation on memory performance, which have already been reported for 30/32 subjects in the current report, but rather in order to assess the relationships between effects of stimulation on fMRI connectivity and on memory performance.

We identified associations between changes in memory performance and task-dependent stimulation effects on global correlativity for each group (individual group model interaction effects) (*Figure 5*). We focused on the a priori hypothesis that the left medial temporal lobe would show the greatest effects based on previous findings of left-lateralized effects of PMN-targeted stimulation on hippocampal fMRI activity (*Kim et al., 2018*; *Nilakantan et al., 2019*) and on the general importance of medial temporal lobe for memory (*Scoville and Milner, 1957*; *Squire, 1992*). For each of the interaction effect clusters within the left medial temporal lobe for both PMN-targeted and PFC-targeted stimulation conditions, a robust linear model was built regressing the interaction effect for that cluster (retrieval task stimulation effect minus resting-state stimulation effect) and tSNR onto the stimulation effect on memory performance (post-stimulation minus post-sham) (R packages *MASS* and *sfsmisc*, RStudio 1.1.453).

## Acknowledgements

This project was funded by R01MH106512 from the National Institute of Mental Health. The content is solely the responsibility of the authors and does not necessarily represent the official views of the National Institutes of Health. The authors would like to thank Stephen VanHaerents for his assistance with TMS safety screening, Sungshin Kim for his contributions to the episodic memory task, and Jonathan O'Neil, Robert Palumbo, Elise Gagnon, Melissa McSweeney, and Melissa Gunlogson for their roles in data collection. This research was supported in part through the computational resources and staff contributions provided for the Quest high-performance computing facility at Northwestern University, which is jointly supported by the Office of the Provost, the Office for Research, and Northwestern University Information Technology. Neuroimaging was performed at the Northwestern University Center for Translational Imaging, supported by Northwestern University Department of Radiology.

## Additional information

### Funding

| Funder | Grant reference number | Author |
|---|---|---|
| National Institute of Mental Health | R01MH106512 | Joel L Voss |
| National Institute of Neurological Disorders and Stroke | T32 NS047987 | Kristen N Warren Molly S Hermiller |

The funders had no role in study design, data collection and interpretation, or the decision to submit the work for publication.

### Author contributions

Kristen N Warren, Data curation, Formal analysis, Investigation, Visualization, Writing—original draft, Writing—review and editing; Molly S Hermiller, Aneesha S Nilakantan, Validation, Investigation, Writing—review and editing; Joel L Voss, Conceptualization, Resources, Supervision, Funding acquisition, Validation, Methodology, Project administration, Writing—review and editing

### Author ORCIDs

Kristen N Warren https://orcid.org/0000-0002-6960-8466
Molly S Hermiller http://orcid.org/0000-0001-7234-4695

### Ethics

Human subjects: The Institutional Review Board at Northwestern University provided approval for this study (STU00070522). All participants gave written, informed consent and were paid for their participation.

### Decision letter and Author response

Decision letter https://doi.org/10.7554/eLife.49458.023
Author response https://doi.org/10.7554/eLife.49458.024

## Additional files

### Supplementary files

• Supplementary file 1. Effect tables for the linear mixed models used to identify the demand-specific effects for PMN-targeted versus PFC-targeted control stimulation (*Figure 1*).
DOI: https://doi.org/10.7554/eLife.49458.017

• Supplementary file 2. Expanded names for PMN and ATN region abbreviations used in *Figure 2*. Region labels from Eickhoff-Zilles macro labels from N27 in MNI space. Note that 'calcarine gyrus' refers to the area surrounding the calcarine sulcus, including the precuneus and lingual gyrus.

DOI: https://doi.org/10.7554/eLife.49458.018

• Supplementary file 3. 18 supra-threshold regions identified via significant interaction between stimulation condition and demand (bolded), followed by their drivers—regions showing significant interaction between stimulation condition and demand in their seed-based connectivity to one of the 18 supra-threshold regions. Region labels from Eickhoff-Zilles macro labels from N27 in MNI space. Note that 'calcarine gyrus' refers to the area surrounding the calcarine sulcus, including the precuneus and lingual gyrus.

DOI: https://doi.org/10.7554/eLife.49458.019

• Supplementary file 4. 15 supra-threshold regions identified via significant interaction between stimulation condition and demand (bolded), followed by their drivers—regions showing significant interaction between stimulation condition and demand in their seed-based connectivity to one of the 15 supra-threshold regions. Region labels from Eickhoff-Zilles macro labels from N27 in MNI space. Note that 'calcarine gyrus' refers to the area surrounding the calcarine sulcus, including the precuneus and lingual gyrus.

DOI: https://doi.org/10.7554/eLife.49458.020

• Transparent reporting form DOI: https://doi.org/10.7554/eLife.49458.021

## Data availability

Raw data are available on the Northwestern University Neuroimaging Data Archive (https://nunda.northwestern.edu/). Access to NUNDA requires creating a login with a valid email address - registration is free and unvetted. Once a login has been created, you can find a project with ID "DERVISH". From there you can find the following subject IDs which were used in the current study. PMN-Stim, post stimulation: 01DY_A2, 02DY_A5, 03DY_A5, 04DY_A2, 05aDY_A2, 06DY_A5, 07aDY_A2, 08DY_A5, 09DY_A2, 10DY_A5, 11aDY_A2, 12aDY_A5, 13DY_A2, 14DY_A5, 15DY_A52 16DY_A25 PMN-Stim, post sham: 01DY_A5, 02DY_A2, 03DY_A2, 04DY_A5, 05aDY_A5, 06DY_A2, 07aDY_A5, 08DY_A2, 09DY_A5, 10DY_A2, 11aDY_A5, 12aDY_A2, 13DY_A5, 14DY_A2, 15DY_A5, 16DY_A2. PFC-Stim, post stimulation: 17DY_A2, 18DY_A5, 19DY_A2, 20DY_A5, 21DY_A2, 22aDY_A5, 23DY_A2, 24DY_A5, 25DY_A2, 26DY_A5, 27DY_A2, 28DY_A5, 29DY_A2, 30DY_A5, 31bDY_A2, 32aDY_A5. PFC-Stim, post sham: 17DY_A5, 18DY_A2, 19DY_A5, 20DY_A2, 21DY_A5, 22aDY_A2, 23DY_A5, 24DY_A2, 25DY_A5, 26DY_A2, 27DY_A5, 28DY_A2, 29DY_A5, 30DY_A2, 31bDY_A5, 32aDY_A2.

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
