## [Decision Letter]

Thank you for submitting your article "Increased task-dependent functional connectivity by stimulation of the hippocampal network predicts memory enhancement" for consideration by *eLife*. Your article has been reviewed by two peer reviewers, and the evaluation has been overseen by a Reviewing Editor and Laura Colgin as the Senior Editor. The reviewers have opted to remain anonymous.

The reviewers have discussed the reviews with one another and the Reviewing Editor has drafted this decision to help you prepare a revised submission.

Summary:

Warren and colleagues report an fMRI study investigating the influence of repetitive TMS on network connectivity during autobiographical retrieval and rest. They found that stimulation of the lateral parietal cortex influenced network connectivity (especially in the posterior medial network) during retrieval more so than in rest, and that hippocampal connectivity changes were correlated with episodic memory improvements in a separate task. The reviewers were all in agreement that the premise behind the paper – determining the impact of TMS on task-dependent interactions – is a very good one, and that the results have the potential for broad impact. However, several substantial concerns were raised. These are listed below.

Essential revisions:

1) The reviewers highlighted a number of analytical issues, which could potentially confound interpretation:

- Sham stimulation was performed at a low intensity, rather than using a form of sham coils. One reviewer is concerned that this would allow participants to identify the sham condition. This reviewer also noted that the sham condition "is ordered with the real week, such that there is no washout period. The authors ideally should perform additional control analysis to confirm the sham is doing what they think it is doing: the order effect should be assessed, does the first week show a response that the second week doesn't show (regardless of TMS condition real/sham). A blinding should have been assessed: e.g. do participants identify the sham successfully – which is especially important given the 'within-subjects' design of real and sham."

- Although change in memory performance was assessed according to the pre-post design, it was not clear if change in connectivity is also assessed in such a fashion. The primary comparison is post-stimulation real to post-stimulation sham, which is only effective if the sham stimulation is effective. A reviewer noted that "the authors cite their own work as justification for this practice, yet none of the cited papers perform any analytical justification for this procedure." More justification should be provided here.

- The site of stimulation was performed in different groups, yet the primary analysis presented was the difference of differences (real-sham parietal – real-sham frontal). The reviewers believe that it is important to show the primary effects (e.g. sham, real parietal, without differences). In addition, the current presentation makes interpreting the bar graphs challenging – what is the standard error being used to assess the differences?

- The main comparisons are between the retrieval task and rest, but without another task condition, it is unknown whether the current results are specific to autobiographical retrieval or whether the same effects would have been observed with any type of task or stimuli.

These concerns led one reviewer to conclude that the "primary issue with this paper is in the presentation, a number of alternative explanations of the effects were not explored, and were presented in such a fashion that would preclude the assessment of those details."

2) All reviewers believe that it is important to demonstrate the network targeted by the dorsolateral prefrontal cortex stimulation, because it could be that a lack of effect from this site that is driving some of the effects observed.

One reviewer noted found this point to be the most central concern: "Probably most concerning is the argument that the memory network was enhanced, but no consideration was given to the lack of an effect on the stimulation site. E.g. one would expect that the most prominent effect would be stimulation on parietal cortex lead to a change in parietal-hippocampal connectivity; and that no effect of dlpfc-hippocampal connectivity was observed." Did control stimulation result in any connectivity changes?

On this point, another reviewer noted "the authors chose two stimulation sites: one in the lateral parietal cortex (used in their previous work) and one in the dorsolateral prefrontal cortex. Both sites were chosen for their functional connectivity and meta-analytic co-activation with the hippocampus. Yet the prefrontal site is referred to as "out of network." This is confusing because the authors generally refer to a broad hippocampal-cortical network that can be parsed into different subnetworks (including the PMN, which includes the parietal site but not the prefrontal site). In that sense, the prefrontal site is certainly "in" a hippocampal-cortical network, but outside of the PMN. As such, one might have reasonably expected that prefrontal stimulation would drive functional connectivity with the hippocampus, but perhaps with a different set of associated targets or with different behavioral consequences. The authors should clarify their logic on this point-- did they expect any effect of prefrontal stimulation? If not, why not, given the way it was selected? Narrowing the focus on the PMN (in- or out- of the PMN) might be helpful. As a related aside, the literature review in the Introduction often refers to changes "in the network" but this could mean almost anything in the current context-- it would be better to be specific about which regions are involved."

3) Third, the authors discuss another network, the anterior temporal network (ATN), which they and others have compared against the functions of the PMN. Based on the initial set of analyses, I might have expected the authors to target an AT site to test for a double dissociation, which would have provided a stronger test of functional specificity. Here's where there is some gray area-- the ATN ROIs used here do include dorsolateral PFC nodes, despite this region not being part of what is typically defined as the ATN. These nodes are in a slightly different location than the stimulation site, coming from studies comparing functional connectivity of the PRC (ATN) and PHC (PMN) (Libby et al., 2012, Ritchey et al., 2014), and in the latter study, the dorsolateral PFC nodes were connected to both the ATN and PMN. It's debatable whether these PFC nodes should be grouped as part of the ATN (I can see arguments both ways), but if they are, then the authors should reflect on how this impacts the interpretation of the prefrontal stimulation condition and the ROI results.

4) The data were analyzed with linear mixed effects models. Such models are typically used when there are multiple observations per bin (e.g., in a hierarchical design with trials nested within subjects), but here there is only one observation per bin and subjects are the only random effect. Why then did the authors choose to use this analysis approach versus a more conventional repeated-measures ANOVA? This did not strike the reviewers as wrong per se, just unusual -- and we would like to know whether the results are similar when using ANOVA.

---

## [Author Response]

Essential revisions:1) The reviewers highlighted a number of analytical issues, which could potentially confound interpretation:- Sham stimulation was performed at a low intensity, rather than using a form of sham coils. One reviewer is concerned that this would allow participants to identify the sham condition. This reviewer also noted that the sham condition "is ordered with the real week, such that there is no washout period. The authors ideally should perform additional control analysis to confirm the sham is doing what they think it is doing: the order effect should be assessed, does the first week show a response that the second week doesn't show (regardless of TMS condition real/sham). A blinding should have been assessed: e.g. do participants identify the sham successfully – which is especially important given the 'within-subjects' design of real and sham."

We now further highlight that there indeed was a considerable washout period between the stimulation week and sham week of, on average, 11.5 weeks (Materials and methods subsection “Experiment Design”). Figure 1 and its caption have been updated to further highlight this important aspect of the experiment design. As indicated, the order of the stimulation and sham weeks was also counterbalanced across subjects, such that significant group-level effects are unlikely to be driven by ordering of conditions. Nonetheless, as suggested we have added new control analyses to test whether the order of the stimulation and sham conditions contributed to the reported effects of stimulation on connectivity. As now described in Materials and methods subsection “fMRI Data Analysis”, we ran an additional linear model that included the order of stimulation/sham as a factor. We found that order did not have a significant main effect on connectivity in either network (PMN: T(83)=-1.6, p=0.12; ATN:T(83)=-0.44, p=0.66) and did not show any interaction with stimulation condition in either network (PMN: T(83)=1.29, p=0.20; ATN:T(83)=-0.30, p=0.77). As now described, the order of the stimulation and sham conditions thus had no significant influences on the effects of stimulation on connectivity that are reported.

We also now further clarify the logic of the experiment design, whereby we compare the effects of active stimulation to sham within the group receiving PMN-targeted stimulation (of parietal cortex) to the group receiving PFC-targeted stimulation (of prefrontal cortex). Although the reviewers suggest that a stronger design could have been achieved by attempting to blind subjects to stimulation versus sham conditions, it is our opinion that there is no perfect sham intensity condition for noninvasive brain stimulation given that there is always the possibility that subjects could experience subtle subjective differences between active and sham conditions. For this reason, we utilize a control group, whereby subjects receive the active and sham intensity conditions but applied to an area (prefrontal cortex) that is not expected to produce the same effects on the hippocampal posterior-medial network. Thus, awareness of the active versus sham conditions alone could not produce the reported effects because this awareness was similar for the two groups whereas hypothesized effects of stimulation on posterior-medial network connectivity were specific to the group receiving PMN-targeted (parietal) stimulation. This is now explained in the fourth paragraph of the Introduction section, and in the Materials and methods subsection “Transcranial Magnetic Stimulation (TMS)”. Further evidence for the suitability of the experiment design is described in our response to the third concern from Essential revision #1, below.

- Although change in memory performance was assessed according to the pre-post design, it was not clear if change in connectivity is also assessed in such a fashion. The primary comparison is post-stimulation real to post-stimulation sham, which is only effective if the sham stimulation is effective. A reviewer noted that "the authors cite their own work as justification for this practice, yet none of the cited papers perform any analytical justification for this procedure." More justification should be provided here.

To clarify, change in memory performance was not assessed using a pre-post comparison, but rather was analyzed as post-stimulation real versus post-stimulation sham, as was also done for the fMRI connectivity analysis. This is indicated in the Materials and methods section (subsection “Episodic memory assessment and analysis of its relationship to fMRI connectivity”) and is now further clarified in the Results section (subsection “Increased memory task-dependent connectivity predicts episodic memory improvement”, first paragraph). In response to the reviewer’s concern, we now include an analysis of the baseline values (pre-stimulation real versus pre-sham), which found no significant difference between conditions at baseline (T(31)=-1.23,p=0.23) (Results subsection “Increased memory task-dependent connectivity predicts episodic memory improvement”, first paragraph). We did not intend for our citations to previous papers to serve as justification for this analysis strategy, but rather to simply highlight that we have utilized this approach before. As we now clarify (see our first response to point #1), the logic is that sham stimulation involves the same experimental procedures but with stimulation delivered at such a low intensity that it should not directly influence brain function. Therefore, comparison of post-stimulation values for real versus sham stimulation will isolate the effects of stimulation versus the neutral sham baseline, particularly when contrasted with the same stimulation versus sham comparison made in subjects receiving all of the same conditions but with stimulation applied to a control (prefrontal) location. Further support for the validity of the sham condition and experiment design is provided in our response to the third concern from Essential revision #1 and the corresponding updates that have been made to the manuscript.

- The site of stimulation was performed in different groups, yet the primary analysis presented was the difference of differences (real-sham parietal – real-sham frontal). The reviewers believe that it is important to show the primary effects (e.g. sham, real parietal, without differences). In addition, the current presentation makes interpreting the bar graphs challenging – what is the standard error being used to assess the differences?

We now clarify that, as shown in the left panel of Figure 2A, the increase in task-dependent connectivity due to stimulation was significant in the PMN for the group of subjects receiving PMN-Stim (parietal) alone. Thus, even though the primary analysis uses comparison of the PMN-Stim group to the control PFC-Stim group, the effects of stimulation on task-dependent connectivity within the PMN-Stim group alone were significant. This is now clarified along with corresponding statistical tests in the second paragraph of the subsection “Stimulation effects on task-dependent fMRI connectivity within hippocampal networks” and via the inclusion of these within-group statistical tests in Figure 2.

As now further clarified in the figure legends, all standard error bars represent inter-subject variability, and there are therefore no error bars on values that derive from comparisons across groups, as the subjects being measured are different. The only exception is Figure 2C, in which the error bars indicate the standard error of three-way interaction of state by condition by target, and this is now clarified in the figure legend. For comprehensiveness, source data files are now provided for all major comparisons.

- The main comparisons are between the retrieval task and rest, but without another task condition, it is unknown whether the current results are specific to autobiographical retrieval or whether the same effects would have been observed with any type of task or stimuli.

We agree this is an important consideration and have added a new paragraph to the Discussion section (sixth paragraph). We have taken precaution to describe our effects as task-dependent rather than memory-specific for this reason, although we believe there are some compelling arguments for why the effects are likely specific to memory retrieval. In particular, connectivity within particular networks is modulated by network-relevant cognitive functions (e.g., Bertolero, Yeo, and D’Esposito 2015 PNAS; Mennes, et al., 2013; Rissman, et al., 2004; Cole, et al., 2014), and increased connectivity within the networks that we interrogated (PMN and AT) is typically specific to memory. Thus, although we agree that the effects of stimulation on task-dependent connectivity are likely not specific to autobiographical memory retrieval, we do suggest that they are likely specific to memory in general. Of course, this point can only be fully settled via future work including other comparison task conditions, which we now acknowledge.

These concerns led one reviewer to conclude that the "primary issue with this paper is in the presentation, a number of alternative explanations of the effects were not explored, and were presented in such a fashion that would preclude the assessment of those details."

We hope that our additions have now clarified our methodology and results to allow for the full assessment by reviewers. Notably, in addition to modifications of existing content, we have added four entirely new paragraphs to the Discussion section that address potential alternative interpretations, as suggested (sixth to ninth paragraphs).

2) All reviewers believe that it is important to demonstrate the network targeted by the dorsolateral prefrontal cortex stimulation, because it could be that a lack of effect from this site that is driving some of the effects observed.One reviewer noted found this point to be the most central concern: "Probably most concerning is the argument that the memory network was enhanced, but no consideration was given to the lack of an effect on the stimulation site. E.g. one would expect that the most prominent effect would be stimulation on parietal cortex lead to a change in parietal-hippocampal connectivity; and that no effect of dlpfc-hippocampal connectivity was observed." Did control stimulation result in any connectivity changes?

As indicated above in our response to the third concern from Essential revision #1, the effects of stimulation on task-dependent connectivity were significant when considering only the group of subjects receiving parietal stimulation to target the PMN (left panel of Figure 2A and corresponding statistical descriptions in the text). Therefore, the main effects reported do not require comparison to any effects (or lack of effects) that resulted from stimulation of the dorsolateral prefrontal cortex in the control group. Furthermore, as indicated in multiple sections of the Results, PFC-Stim did not result in lack of effects on connectivity, as suggested by the reviewers. Rather, PFC-Stim significantly decreased task-dependent connectivity in several regions (Figure 4B). This is now clarified in the text (Results subsection “Exploratory whole-brain analysis of task-dependent effects of PMN-targeted stimulation”, fourth paragraph and subsection “Increased memory task-dependent connectivity predicts episodic memory improvement”, last paragraph, Discussion, sixth paragraph). Furthermore, we now note in the Discussion (seventh paragraph) that this pattern is consistent with a previous report of the same PFC-Stim control condition, in which we found that PFC-Stim reduced fMRI correlates of memory encoding within the prefrontal cortex (Kim et al., 2018). Thus, contrary to the effects of PMN-targeted stimulation, the control PFC-Stim condition seems to disrupt fMRI connectivity related to memory, although there were no significant adverse effects on memory performance (subsection “Increased memory task-dependent connectivity predicts episodic memory improvement”, first paragraph).

Although it may be surprising that stimulation of the parietal cortex in the main group (PMN-Stim) did not significantly affect parietal-hippocampal connectivity, this finding is consistent with our previous results using the current multi-day stimulation protocol. In multiple studies we found that the effects of stimulation on fMRI connectivity and memory-related activity were robust for the hippocampus and the medial aspects of the posterior-medial network, with weak or null effects on connectivity and activity of the parietal area that was directly stimulated (Wang, et al., 2014; Kim et al. 2018; Nilakantan et al., 2019; Freedberg et al.,in press). We have included two new Discussion paragraphs on this surprising yet consistent aspect of the findings (eighth and ninth paragraphs).We speculate that because the lateral parietal cortex is one of the more weakly connected portions of the hippocampal network (Kahn et al. 2008), separated by at least two synapses from the mesial temporal lobe (Pandya and Seltzer, 1982; Mesulam et al., 1977), it can serve as a conduit by which noninvasive stimulation can influence the network, but is not affected as much as areas that are more central components of the network with higher levels of functional connectivity and more direct anatomical connectivity with the hippocampus (Pandya et al. 1981, Experimental Brain Research; Kahn et al. 2008, J Neurophys). Indeed, we have found that the degree to which a region’s functional connectivity increases due to stimulation is predicted by its connectivity with the hippocampus (Wang, et al., 2014), and this relationship has been replicated by an independent group (Freedberg et al.,In Press).

On this point, another reviewer noted "the authors chose two stimulation sites: one in the lateral parietal cortex (used in their previous work) and one in the dorsolateral prefrontal cortex. Both sites were chosen for their functional connectivity and meta-analytic co-activation with the hippocampus. Yet the prefrontal site is referred to as "out of network." This is confusing because the authors generally refer to a broad hippocampal-cortical network that can be parsed into different subnetworks (including the PMN, which includes the parietal site but not the prefrontal site). In that sense, the prefrontal site is certainly "in" a hippocampal-cortical network, but outside of the PMN. As such, one might have reasonably expected that prefrontal stimulation would drive functional connectivity with the hippocampus, but perhaps with a different set of associated targets or with different behavioral consequences. The authors should clarify their logic on this point-- did they expect any effect of prefrontal stimulation? If not, why not, given the way it was selected? Narrowing the focus on the PMN (in- or out- of the PMN) might be helpful. As a related aside, the literature review in the Introduction often refers to changes "in the network" but this could mean almost anything in the current context-- it would be better to be specific about which regions are involved."

We now clarify that only the parietal site could be chosen based solely on its functional connectivity with the hippocampus, whereas the control prefrontal site had to be selected via an approach involving meta-analytic co-activation (Materials and methods subsection “Identification of stimulation locations”, last paragraph). This is an important point, as multiple studies have found that dorsolateral prefrontal cortex has weak functional connectivity with the hippocampus and therefore is not typically considered within its resting-state functional connectivity network (Kahn, et al., 2008; Yeo, et al., 2011). Thus, although dorsolateral prefrontal cortex supports memory and is typically active when the hippocampus is active (meta-analytic co-activation), these areas typically have low functional connectivity and therefore the prefrontal cortex serves as a control location under the hypothesis that only stimulation of functionally connected regions will influence the hippocampus.

In addition, we now clarify that the PMN was the network of interest and that the PFC control location was “out of network” with respect to the PMN. We have updated the title of the manuscript to reflect this focus. We have also devoted a new portion of the Discussion section to the possibility that prefrontal cortex could be a component of the broader hippocampal network, but that clearly the effects of stimulation differed significantly for PMN-Stim versus PFC-Stim, with only PMN-Stim significantly increasing task-dependent connectivity of the PMN and improving memory performance, as predicted (Discussion, seventh paragraph).

As suggested, we have also narrowed the anatomical definitions of stimulation-responsive regions in the literature review within the Introduction, to clarify our use of the term “in the network” as pertaining particularly to the PMN.

3) Third, the authors discuss another network, the anterior temporal network (ATN), which they and others have compared against the functions of the PMN. Based on the initial set of analyses, I might have expected the authors to target an AT site to test for a double dissociation, which would have provided a stronger test of functional specificity. Here's where there is some gray area-- the ATN ROIs used here do include dorsolateral PFC nodes, despite this region not being part of what is typically defined as the ATN. These nodes are in a slightly different location than the stimulation site, coming from studies comparing functional connectivity of the PRC (ATN) and PHC (PMN) (Libby et al., 2012, Ritchey et al., 2014), and in the latter study, the dorsolateral PFC nodes were connected to both the ATN and PMN. It's debatable whether these PFC nodes should be grouped as part of the ATN (I can see arguments both ways), but if they are, then the authors should reflect on how this impacts the interpretation of the prefrontal stimulation condition and the ROI results.

As the reviewer indicates, the area of PFC that was stimulated does not overlap any nodes of either the ATN or PMN as defined by the previous studies that are referenced. Therefore, we consider it as a suitable control location against which effects of PMN-targeted stimulation can be compared. Full evaluation of whether dorsolateral PFC is or is not part of the hippocampal network is beyond the scope of our study, but we now highlight (Discussion, seventh paragraph) that this is an open question and emphasize that, in either case, the effects of PFC-Stim were statistically independent from those of PMN-Stim. PFC-Stim had the opposite effect on task-dependent connectivity as PMN-Stim within the PMN (left panel of Figure 2A) and had minimal influence on connectivity within the ATN (right panel Figure 2A). Thus, stimulation of dorsolateral PFC had minimal effect on the ATN, whether it is part of the ATN or not.

4) The data were analyzed with linear mixed effects models. Such models are typically used when there are multiple observations per bin (e.g., in a hierarchical design with trials nested within subjects), but here there is only one observation per bin and subjects are the only random effect. Why then did the authors choose to use this analysis approach versus a more conventional repeated-measures ANOVA? This did not strike the reviewers as wrong per se, just unusual -- and we would like to know whether the results are similar when using ANOVA.

Linear mixed effects models implemented in AFNI allow for the same grouping of conditions and factors as RM-ANOVA but with the additional ability to include covariates, which is not possible using AFNI’s RM-ANOVA command. As indicated in the Materials and methods section (subsection “fMRI Data Analysis”, first paragraph), tSNR was used as a covariate to guard against spurious influence of scan data quality on effects attributed to stimulation (see Materials and methods). However, we have performed the same analyses using RM-ANOVA without any covariates and the same overall pattern of results was obtained. We feel that the analyses including the tSNR covariate are advantageous and so retain them in the manuscript.